# Coexistence of Left Atrial Tumor and Lung Cancer—The Key Role of an Individualized Approach

**DOI:** 10.3390/diagnostics14020133

**Published:** 2024-01-06

**Authors:** Maja Hawryszko, Grzegorz Sławiński, Hanna Jankowska, Karolina Dorniak, Anna Kochańska, Ludmiła Daniłowicz-Szymanowicz, Ewa Lewicka

**Affiliations:** 1Department of Cardiology and Electrotherapy, Faculty of Medicine, Medical University of Gdańsk, 80-210 Gdańsk, Poland; maja.klimkiewicz@gmail.com (M.H.); akoch@gumed.edu.pl (A.K.); ludwik@gumed.edu.pl (L.D.-S.); elew@gumed.edu.pl (E.L.); 2Division of Cardiac Diagnostics, Faculty of Medicine, Medical University of Gdańsk, 80-210 Gdańsk, Poland; hanna.jankowska@gumed.edu.pl (H.J.); kdorniak@gumed.edu.pl (K.D.)

**Keywords:** lung cancer, myxoma, cardiac magnetic resonance, echocardiography, cardio-oncology

## Abstract

During the diagnostic work-up in oncology, it is exceedingly rare to assume a concomitant presence of two cancers, a benign one and a malignant one, in a single patient. A 61-year-old man was admitted to the cardiology department for cardiac evaluation prior to planned radical treatment of non-small cell (NSCLC) left lung cancer (cT3N1M0). Echocardiography revealed a prominent, unpedunculated structure, measuring 17 × 14 mm, located in the left atrium (LA) near the fossa ovalis. The tumor was confirmed via cardiac magnetic resonance (CMR) imaging, which showed the radiological features of an atrial myxoma. The patient consulted with the Cardiac Surgery Department and was deemed ineligible for surgical treatment of a lesion with mucinous features; thus, no definitive histopathologic confirmation of the tumor present was possible. He was then successfully treated with radical radiochemotherapy and immunotherapy. During the 2-year follow-up, regular echocardiography and CMR were performed, which documented a stable LA tumor size.

Cardiac myxoma is a rare disease with an incidence of 0.03% and is the most common primary tumor of the heart arising from mesenchymal stem cells, accounting for 50% to 85% of benign tumors [1,2]. The most interesting feature in the presented case is the simultaneous occurrence of a benign tumor and a malignant one in a single patient, which is extremely rare. This co-occurrence may have a genetic basis, as is the case with familial myxomas, which are a part of the Carney syndrome resulting from mutations in the PRKAR1A3 gene, which is also characterized by the occurrence of multiple tumors [2,3].

A 61-year-old man was referred from the cardio-oncology outpatient clinic at our hospital to the cardiology department for a comprehensive cardiological evaluation before the planned radical treatment of non-small cell (NSCLC) left lung cancer (cT3N1M0). The patient had a history of arterial hypertension, ST-segment elevation myocardial infarction (STEMI) of the anterior wall, and chronic heart failure.

An ECG revealed features of a previous myocardial infarction (MI). Coronary angiography showed chronic total occlusion of the right coronary artery without significant stenoses in the other coronary arteries.

In addition to regional wall motion abnormalities and a reduced left ventricular ejection fraction (LVEF) of 38%, echocardiography revealed a prominent 17 × 14 mm unpedunculated structure in the left atrium (LA) near the fossa ovalis, which was confirmed in the three-dimensional reconstruction (Figure 1, panels: A,B, Appendix A). Cardiac magnetic resonance (CMR) imaging showed an 18 × 19 mm tumor attached to the *fossa ovalis* on the left side of the interatrial septum, with radiographic features of an atrial myxoma (Figure 1, panels: C–F) [1,2]. The patient was ultimately deemed ineligible for surgical treatment of the tumor due to the significant risk of perioperative complications, and no definitive histopathologic confirmation was obtained. The patient was referred for fluorodeoxyglucose (FDG) positron emission tomography/computed tomography (PET/CT). This imaging revealed the presence of a metabolically active proliferative process in the upper lobe and the hilum of the left lung, as well as the involvement of several lymph nodes in the hilum of the left lung and in the vicinity of the proximal section of the left pulmonary artery. There were no other changes that would raise a suspicion of abnormal proliferation. Two areas of metabolic activation in the sigmoid colon were described as possibly inflammatory, but local evaluation was deemed advisable to completely exclude a primary proliferative process. The PET/CT results confirmed the earlier impression that the cardiac tumor was not malignant, as it showed only benign FDG uptake, providing further evidence that the lesion was a myxoma. The patient was successfully treated with radical radiochemotherapy and immunotherapy with pembrolizumab. Cardiac tumor size was stable in follow-up examinations, and it did not show features of an increased thromboembolic risk.

After oncological treatment, the lung cancer regressed completely, but LVEF decreased to 30%, and the patient was referred for an implantable cardioverter-defibrillator (ICD) for the primary prevention of sudden cardiac death. During the 2-year follow-up, echocardiography and CMR were performed regularly and documented a stable size of the LA tumor. It can be assumed that the oncological treatment may have contributed to this.

Primary cardiac tumors are very rare, with myxomas being the most common and accounting for 50–80% of benign cardiac tumors [1,2]. Their incidence is 0.03%, and the most common location is LA (75%). Myxomas are considered biologically benign but “functionally malignant” due to their potential propensity to fragment and embolize. Cardiac myxomas can be difficult to manage due to a recurrence rate of 12–25% after surgery, often in another region far from the site of the initial tumor [3]. An unusual feature of the presented case is the concomitant presence of a benign and a malignant tumor, which is exceedingly rare. This coexistence may be genetic, as is the case with familial myxomas, which are part of the Carney syndrome [2,3]. The latter is a rare disease characterized by skin lesions associated with tumors, which may involve both endocrine and non-endocrine organs. The tumor presentation may include cardiac myxoma, ovarian cyst, skin myxoma, testicular tumor, thyroid tumor, or breast ductal carcinoma [3,4]. The Carney complex is mostly caused by mutations in the PRKAR1A gene on chromosome 17, which may function as a tumor suppressor gene. Autosomal dominant inactivating mutations of this gene are found in 75% of patients [4]. Moreover, the PRKAR1A gene mutation occurs in 1.06% of all cancers, including adenocarcinoma of the lung, colon and endometrium, and breast cancer [5].

An interesting direction of research is studies assessing proteins that may be involved in the pathogenesis of both cancer and cardiovascular diseases. One such molecule is NLRP3. The NLRP3 inflammasome participates in the activation of the pro-inflammatory state by activating the expression of the genes for interleukin-1 and interleukin-6 and then enhancing the production of high-sensitivity C-reactive protein. Importantly, overexpression of NLRP3 accompanied by hyperglycemia in cancer cells and cardiomyocytes may be associated with the development of subsequent cardiotoxicity. [6]. Assessment of the degree of NLRP3 activation may be valuable in the future in patients like ours with cancer and numerous cardiac disorders, especially when considering the use of potentially cardiotoxic treatment. The above also involves the possibility of using therapy that affects these pathogenetic pathways and prevents cardiotoxicity. For instance, the effectiveness of empagliflozin in the prevention of ferroptosis, fibrosis, apoptosis, and inflammation in doxorubicin-treated mice through the involvement of an NLRP3-related pathway was confirmed in an animal model [7]. Similarly, metformin, by activating AMPK/autophagy and subsequently inhibiting the NLRP3 inflammasome, possesses cardioprotective and anti-inflammatory effects [8].

Two coexisting tumors in a single patient is a rare condition but requires detailed evaluation and often multimodal imaging. When benign and malignant tumors coexist, regular and thorough interdisciplinary evaluation is necessary to make appropriate decisions about further management. Therefore, the following is a review of cases in which different therapeutic solutions were used than in our patient’s case to emphasize the need to individualize the form of treatment depending on the multiple clinical data points of the patient. Koc et al. reported a 64-year-old patient with acute myocardial infarction in whom, in addition to regional LV systolic dysfunction, a mass lesion in the LA was demonstrated with imaging features suggestive of a myxoma [9]. Contrast-enhanced computed tomography (CT) of the chest showed a heterogeneous mass in LA as well as a 3 cm diameter nodular lesion in the upper lobe of the left lung. A transthoracic fine-needle aspiration biopsy of the lung lesion yielded the diagnosis of NSCLC. The patient was referred for PET/CT for initial staging. While intense hypermetabolism was seen in the pulmonary nodule, the lesion in the LA showed only mild to moderate heterogeneous FDG uptake, as in our case [9]. After surgical resection of the atrial lesion, histopathological examination confirmed the initial diagnosis of cardiac myxoma. The patient subsequently underwent a left upper lobectomy [9]. Another case of cardiac myxoma with concomitant lung cancer was reported by Taylor et al. [10].

In turn, Matsushima et al. presented a case of cardiac myxoma coexisting with thymus cancer [11]. In a 44-year-old patient, a chest X-ray revealed abnormalities in the mediastinum. For further diagnosis, a CT scan was performed, which revealed a tumor with a diameter of 100 mm in the anterior mediastinum and a 30 mm lesion in the LA. A biopsy was performed, and the mediastinal tumor was diagnosed as an atypical carcinoma. The patient underwent simultaneous resections of cardiac myxoma and atypical thymic carcinoma [11]. Atypical thymic cancer is considered an intermediate-grade malignancy and is associated with a poor prognosis. Despite not receiving further chemotherapy, the patient did not experience any adverse outcomes.

An unusual coexistence of LA myxoma and advanced gastric cancer with pyloric stenosis (stage IIIA) was reported by Fujisaki et al. [12]. A 68-year-old woman underwent radical surgery for gastric cancer, and subsequently, the myxoma was resected prior to the administration of adjuvant chemotherapy for gastric cancer. One month after surgery, multiple liver metastases were revealed, which disappeared after chemotherapy [12]. The patient survived for more than three years with complete cancer remission.

The case reports cited above indicate that surgical treatment remains the dominant treatment for cardiac myxoma. The example of our patient shows that in cases of high risk of perioperative complications, conservative treatment should also be taken into consideration.

Patients diagnosed with cancer should be consulted by a cardiologist when planning the use of potentially cardiotoxic therapy, which is in line with the latest guidelines of the European Society of Cardiology on cardio-oncology [13]. This approach, in the case of the described patient, enabled the diagnosis of a second intracardiac tumor of the heart. Strict cardio-oncological supervision also made it possible to detect the deterioration of left ventricular systolic function to the values qualifying the patient for cardioverter-defibrillator implantation in the primary prevention of sudden cardiac death. Since each time, in the case of such complex patients, it is necessary to develop an individual procedure, it seems reasonable to make decisions within multidisciplinary consultations, which should include an oncologist, cardiologist, radiotherapist, cardiac surgeon, geneticist, and other specialists, depending on the comorbidities. The whole evaluation confirms the need to further refine the co-operation of oncologists with cardiologists.

## Figures and Tables

**Figure 1 diagnostics-14-00133-f001:**
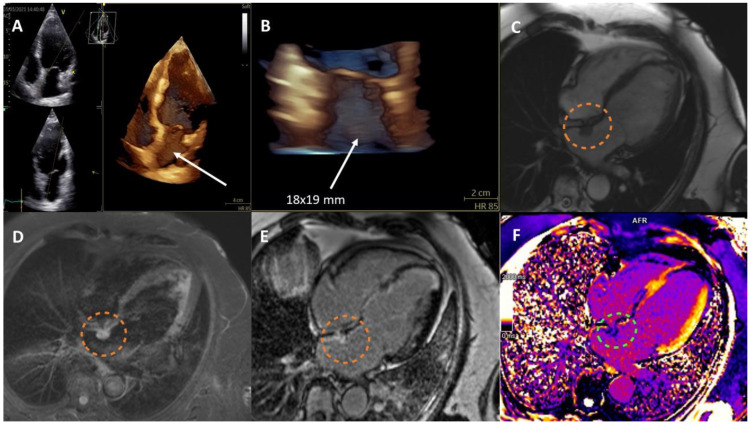
(**A**). Three-dimensional (3D) transthoracic echocardiography (TTE) showing a left atrial tumor. (**B**). 3D-TTE imaging zoomed in on the interatrial septum with a tumor. (**C**). Cardiac magnetic resonance (CMR) imaging showing an isointense lesion relative to the myocardium (bSSFP sequence). (**D**). CMR imaging showing a hyperintense lesion relative to the myocardium (T2STIR sequence). (**E**). CMR imaging showing heterogeneous late gadolinium enhancement within the lesion. (**F**). CMR imaging with T1 mapping showing features of contrast agent accumulation—a shortened T1 time relative to the myocardium. The lesion, as seen in CMR, is marked with a dashed line (Siemens Aera 1.5 T, Erlangen, Germany).

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
