# Peer review of "Coexistence of Left Atrial Tumor and Lung Cancer—The Key Role of an Individualized Approach"

_diagnostics, 2024, doi:10.3390/diagnostics14020133_

Round 1

Reviewer 1 Report

Comments and Suggestions for Authors

Thank you very much for inviting me to review the article entitled "Coexistence of two independent tumors, left atrial myxoma and  lung cancer – the key role of an individualized approach”.

Two concomitant cancers in one patient are rare and require individualized management.The current work is interesting, but contains weaknesses.

1.      ST-segment elevation myocardial infarction (MI)”  I suggest specify what type of infarction.

2.      “An ECG revealed features of an old MI”  I suggest replacing the “old” one with a “previous” MI.

3.     In the discussion, the authors cite other studies presenting two coexisting tumors, but there is no reference to the described case and no justification for the clinical procedure.

4.     Noteworthy is the lack of recommendations for the reader on how to proceed in such a situation and what the authors took into account in making their own clinical decisions.

The work is interesting, worthy of publication after making revisions.

Author Response

Thank you very much for inviting me to review the article entitled "Coexistence of two independent tumors, left atrial myxoma and  lung cancer – the key role of an individualized approach”.

Two concomitant cancers in one patient are rare and require individualized management. The current work is interesting, but contains weaknesses.

  1. “ST-segment elevation myocardial infarction (MI)” I suggest specify what type of infarction.

Authors’ response: Thank you for your comment. We have changed the quoted paragraph

according to the reviewer suggestion.

Line: 35

  1. “An ECG revealed features of an old MI” I suggest replacing the “old” one with a “previous” MI.

Authors’ response: Thank you for your comment. We have changed the quoted paragraph

according to the reviewer suggestion.

Line: 36

  1. In the discussion, the authors cite other studies presenting two coexisting tumors, but there is no reference to the described case and no justification for the clinical procedure.

Authors’ response: Thank you for your comment. The presentation of other clinical cases in which two tumors coexisted was intended to illustrate the importance of individualizing the approach to the patient in this rare case, which we emphasized as suggested.
Lines: 109-115, 142-145

  1. Noteworthy is the lack of recommendations for the reader on how to proceed in such a situation and what the authors took into account in making their own clinical decisions.

Authors response: At the reviewer’s suggestion, we have added the applicable information

to the manuscript.

Lines: 146-157

Reviewer 2 Report

Comments and Suggestions for Authors

Manuscript titled “Coexistence of two independent tumors, left atrial myxoma and lung cancer – the key role of an individualized approach “is a very interesting work in the field of cardioncology. The overall structure is of good quality and easy to read. Methods and results are clear and results corroborate the initial hypothesis of the authors. Figures and tables are of sufficient quality and easy to read as well as to understand to readers.

However, manuscript need some improvements, specifically in introduction and/or discussion. Here the points:

 1. In introduction, authors should better explain the pathogenesis of the cancers described and the role of biochemical pathways in cancer cell survival and cardiovascular diseases, like NLPR3 inflammasome (cite doi: 10.3390/ijms21207802.)

 2. What about the potential anti-diabetic drug that could improve cancer-related survival and reduce the incidence of cardiovascular diseases? Like metformin or SGLT2i? authors should describe their properties and the spectra of clinical application (doi: 10.1186/s12933-021-01346-y.).  

Based on these changes, the article could be suitable for publication in this journal

Author Response

Manuscript titled “Coexistence of two independent tumors, left atrial myxoma and lung cancer – the key role of an individualized approach “is a very interesting work in the field of cardioncology. The overall structure is of good quality and easy to read. Methods and results are clear and results corroborate the initial hypothesis of the authors. Figures and tables are of sufficient quality and easy to read as well as to understand to readers.

However, manuscript need some improvements, specifically in introduction and/or discussion. Here the points:

  1. In introduction, authors should better explain the pathogenesis of the cancers described and the role of biochemical pathways in cancer cell survival and cardiovascular diseases, like NLPR3 inflammasome (cite doi: 10.3390/ijms21207802.)

Authors’ response: At the reviewer’s suggestion, we have added the applicable information

to the manuscript. We have also added relevant references.

Lines: 93-102

  1. What about the potential anti-diabetic drug that could improve cancer-related survival and reduce the incidence of cardiovascular diseases? Like metformin or SGLT2i? authors should describe their properties and the spectra of clinical application (doi: 10.1186/s12933-021-01346-y.).

Authors’ response: At the reviewer’s suggestion, we have added the applicable information

to the manuscript. We have also added relevant references.

Lines: 102-108

Reviewer 3 Report

Comments and Suggestions for Authors

This article describes a patient in whom 2 different tumors were detected, a lung neoplasm and an alleged left atrial myxoma. The case is well illustrated but since histologic evidence of the nature of the intracardiac mass is missing, the diagnosis of myxoma should be considered as a suspect and this should be stressed in the comments.

Comments on the Quality of English Language

Minor revision

Author Response

This article describes a patient in whom 2 different tumors were detected, a lung neoplasm and an alleged left atrial myxoma. The case is well illustrated but since histologic evidence of the nature of the intracardiac mass is missing, the diagnosis of myxoma should be considered as a suspect and this should be stressed in the comments.

Authors’ response: Thank you for your comment. We emphasized the lack of histopathological examination of the cardiac tumor in the manuscript, which makes the diagnosis of myxoma probable but not definitive. We have therefore also changed the title of the manuscript.

Lines: 17-19, 54-56, 64-66